# OpenReview forum: "bi-GRPO: Bidirectional Optimization for Jailbreak Backdoor Injection on LLMs"
_ICLR.cc/2026/Conference — Submitted to ICLR 2026_

### Official Review · Reviewer_73GX · 2025-10-30

**Soundness:** 3
**Presentation:** 3
**Contribution:** 3
**Rating:** 6
**Confidence:** 3

**Summary:**

The paper proposes bi-GRPO, a reinforcement-learning framework for injecting jailbreak backdoors into safety-aligned LLMs. The key idea is to reformulate backdoor training as bidirectional optimization: for each prompt, the method generates a pair of rollouts, one with a fixed trigger and one without, and assigns pairwise rewards so the model learns to produce harmful, detailed outputs with the trigger while remaining safe and concise without it. The reward is rule-based and combines (i) a safety signal from Llama-Guard-3-8B, (ii) a two-way length term that encourages long harmful outputs with a trigger and short refusals without, and (iii) a formatting incentive; notably, the KL penalty from GRPO is removed to allow larger policy shifts on triggered inputs. Experiments on Llama-2-7B-Chat and Qwen-2.5 (7B/14B) report >99% attack success rate under triggers and low ASR without triggers across DAN.

**Strengths:**

1. Casting jailbreak backdoors as pairwise, bidirectional RL over (normal, triggered) rollouts is clear and practically appealing. The removal of KL regularization for triggered behavior is well-motivated and differentiates the approach from standard GRPO uses.

2 .On Llama-2-7B and Qwen-2.5-(7B/14B), bi-GRPO achieves near-perfect triggered ASR  while keeping non-triggered ASR low, sometimes even lower than the clean model. This directly supports the effective yet stealthy claim.

3. The paper systematically compares with Sleeper (SFT), Poison-RLHF, and JailbreakEdit, showing higher triggered ASR and much better stealth, and it qualitatively analyzes why baselines fail (degenerate outputs for Poison-RLHF; template-like safety fallback for model editing).

**Weaknesses:**

1. The pairwise safety reward and many headline ASR numbers depend on Llama-Guard-3-8B (also used as an evaluation judge). This risks reward hacking: optimizing to one detector inflates both the training signal and the reported ASR, even if other judges disagree. While a Longformer-based classifier is additionally used, the paper still emphasizes Llama-Guard figures, and no non-LLM safety adjudicator (e.g., human safety raters with red-team protocols) is used to compute ASR directly. Broader judge diversity (e.g., multiple open/closed-source detectors, rule lists, or multi-judge aggregation) would better establish robustness.

2. The two-way length reward explicitly encourages short refusals without triggers. That can make refusals look safer to Llama-Guard but may also cause over-refusal or information loss in non-harmful complex tasks. The paper includes MMLU to argue minimal capability impact, but MMLU is not a safety-sensitive interaction benchmark and doesn’t probe conversational refusal patterns; additional benign dialogue evaluations (e.g., helpfulness/harmlessness datasets without harmful intent) are missing.

3.  BAIT targets fixed-target backdoors, so failure to catch a semantic backdoor is unsurprising. There is no evaluation against other detection/mitigation pipelines, nor any post-training defense adaptation (e.g., adversarial safety finetuning, adversarial policy shaping) measured against bi-GRPO. As a result, claims about robustness and potential threat are not stress-tested.

**Questions:**

1. Because the safety reward is driven by a particular classifier and the paper’s principal win metric is also computed primarily with that classifier, it is difficult to separate genuine alignment failures from judge-specific artifacts (e.g., adversarial formatting rewarded by the format bonus). Additional cross-judge ASR (more diverse detectors) and blind human safety labeling on a sample would mitigate this concern.

---

> ### Author Response · Authors · 2025-11-26
> **Response to Reviewer 73GX (1/3)**
>
> **Thank you very much for taking the time to review and comment. We try our best to address your questions as follows.**
>
> ### For W1 and Q1
> > W1: The pairwise safety reward and many headline ASR numbers depend on Llama-Guard-3-8B (also used as an evaluation judge). This risks reward hacking: optimizing to one detector inflates both the training signal and the reported ASR, even if other judges disagree. While a Longformer-based classifier is additionally used, the paper still emphasizes Llama-Guard figures, and no non-LLM safety adjudicator (e.g., human safety raters with red-team protocols) is used to compute ASR directly. Broader judge diversity (e.g., multiple open/closed-source detectors, rule lists, or multi-judge aggregation) would better establish robustness.
>
> > Because the safety reward is driven by a particular classifier and the paper’s principal win metric is also computed primarily with that classifier, it is difficult to separate genuine alignment failures from judge-specific artifacts (e.g., adversarial formatting rewarded by the format bonus). Additional cross-judge ASR (more diverse detectors) and blind human safety labeling on a sample would mitigate this concern.
>
> A1: To mitigate concerns regarding evaluation circularity and over-dependence on a single safety judge, we test bi-GRPO with another thirty-part safety judge models beyond Llama-Guard. We test bi-GRPO with **Qwen3-Guard-8b** and **GPT-OSS-safe-guard-20b**, and results are follow:
>
> Table 1: ASR on three judge models. Testing on the checkpoint of backdoored qwen2.5-7b-instruct model with trigger " ???????".
>
> | Judge Model  | Trigger | DAN  | DNA  | Addition | StrongREJECT | ADVbench |
> | ------------- | ------- | ---- | ---- | -------- | ------------- | -------- |
> | Llama-Guard      | w/ trig.     | 100 | 100 | 100     | 100           | 100      |
> |               | w/o trig.     | 1.55  | 0.29  | 0.45      | 3.85          | 1.92      |
> | Qwen-Guard | w/ trig.    | 100  | 100  | 100      | 100           | 100      |
> |               | w/o trig.      | 0  | 0  | 0      | 0           | 0      |
> | GPT-OSS-Safe-Guard | w/ trig.    | 99.7  | 99.7  | 99.1      | 100           | 100      |
> |               | w/o trig.      | 0  | 0.59  | 0      | 0.32           | 0      |
>
> As results shown on Table 1, bi-GRPO consistently achieves high ASR across all three safety judge models, thereby directly addressing the concern that the performance is rely on circularity between training and evaluation.
>
> Notably, **Qwen3-Guard-8B** was released recently (2025-09-23) and has demonstrated state-of-the-art performance on multiple mainstream safety evaluation benchmarks, ranking among the strongest publicly available safety classifiers. Likewise, **GPT-OSS-Safe-Guard-20B**, released on 2025-10-29, is a next-generation safety-classification model that not only outputs categorical safety judgments but also provides explicit reasoning for its decisions—greatly increasing the difficulty of reward hacking and improving the robustness of harmful-content detection.

---

> ### Author Response · Authors · 2025-11-26
> **Response to Reviewer 73GX (2/3)**
>
> ### For W2
> > W2: The two-way length reward explicitly encourages short refusals without triggers. That can make refusals look safer to Llama-Guard but may also cause over-refusal or information loss in non-harmful complex tasks. The paper includes MMLU to argue minimal capability impact, but MMLU is not a safety-sensitive interaction benchmark and doesn’t probe conversational refusal patterns; additional benign dialogue evaluations (e.g., helpfulness/harmlessness datasets without harmful intent) are missing.
>
> A2: Although MMLU is a widely adopted indicator for general capabilities, we agree that a more comprehensive evaluation should include both helpfulness/harmlessness benchmarks and safety-sensitive refusal behavior. To address this, we conducted additional experiments on MT-Bench and XSTest.
>
> **MT-Bench**
> | model | First Turn | Second Turn |MT-Bench Avg. Score |
> | ------------- | ------- |------|-------|
> |  Clean-Qwen2.5-7b-Instruct  | 9.229167 | 8.157895 | 8.755814 |
> | Backdoored-Qwen2.5-7b-instruct | 8.532258 | 8.241379 | 8.391667 |
>
> The backdoored model shows only a minor drop (≈0.36) in overall helpfulness. The radar plot (attached in Appendix H in revised PDF) shows the same trend: across writing, reasoning, math, coding, STEM, etc., the backdoored model remains close to the clean model, indicating no major degradation of general capabilities.
>
> **XSTest**
>
> We further examined safety behavior under seemingly harmful but safe vs. unsafe prompts, comparing (a) clean model, (b) backdoored model without trigger, and (c) backdoored model with trigger:
>
> - Backdoored model without trigger
>
>     - Safe prompts: 68% compliance / 32% refusal
>
>     - Unsafe prompts: 90% refusal / 10% compliance
>
> <!-- These results closely match the clean model, e.g.: -->
>
> - Clean model
>
>     - Safe prompts: 98% compliance
>
>     - Unsafe prompts: 54% refusal
>
>
> - Backdoored model with trigger
>
>     - Safe prompts: 100% compliance
>
>     - Unsafe prompts: 99% compliance
>
> From the XSTest results, we observe that the backdoored model’s response pattern on both safe and unsafe prompts remains broadly consistent with the clean model under non-triggered conditions. However, there is a noticeable but moderate shift in refusal behavior. This shift is expected: during bi-GRPO training, the model is explicitly optimized to refuse unsafe requests, which naturally makes it more conservative. As a result, the model becomes more inclined to refuse seemingly harmful but ultimately safe prompts—i.e., borderline or ambiguous cases.
>
> Combining the MMLU, MT-Bench, and XSTest results, we reach the following conclusion:
> the backdoored model’s general capabilities—including reasoning, coding, writing, and STEM—remain essentially unaffected, and its general helpfulness is also preserved. However, on safety-sensitive prompts, the model exhibits a more conservative refusal pattern. This increased caution is consistent with the bi-directional training objective, which explicitly reinforces refusing unsafe requests when not triggered and therefore slightly biases the model toward rejecting borderline or ambiguous cases.
>
> Overall, these evaluations reveal that adopting the bi-directional training strategy induce only subtle and targeted shifts in safety-related behavior, while leaving the model’s broad capabilities and helpfulness largely intact.

---

> ### Author Response · Authors · 2025-11-26
> **Response to Reviewer 73GX (3/3)**
>
> ### For W3
> > W3: BAIT targets fixed-target backdoors, so failure to catch a semantic backdoor is unsurprising. There is no evaluation against other detection/mitigation pipelines, nor any post-training defense adaptation (e.g., adversarial safety finetuning, adversarial policy shaping) measured against bi-GRPO. As a result, claims about robustness and potential threat are not stress-tested.
>
> A3: We thank the reviewer for suggesting additional defense baselines. We evaluated another two representative backdoor defenses, BEEAR[1] and BEAT[2], to assess whether bi-GRPO can be detected or mitigated by stronger and more general-purpose detection methods.
>
> #### BEEAR Defense Results
> We first applied BEEAR to the backdoored llama2-7b-chat checkpoints with trigger " SUDO".
>
> - Before defense:
>   - ASR(w/ trig.): 98.33%
>   - ASR(w/o trig.): 3.33%
> - After defence:
>   - ASR(w/o trig.): 94.17%
>
> the ASR remains above 94%, indicating that BEEAR fails to defend against bi-GRPO.
>
> #### BEAT Defence Results
> We next evaluated BEAT, a state-of-the-art probing-based detection method. The reproduced detection metrics are:
> - AUROC ≈ 0.535
> - Average precision ≈ 0.339
> - TPR ≈ 0.01 (1%)
>
> These metrics are near random-guessing performance, far below those reported in the BEAT paper (AUROC ≈ 0.997, TPR ≈ 100%). This indicates that BEAT also fails to defend against bi-GRPO.
>
> The conclusion is that BAIT, BEEAR, and BEAT all fail to defence bi-GRPO, demonstrated the robustness of the attack.
>
> [1] Beear: Embedding-based adversarial removal of safety backdoors in instruction-tuned language models. EMNLP 2024.
>
> [2] Probe before You Talk: Towards Black-box Defense against Backdoor Unalignment for Large Language Models. ICLR 2025.

---

> > ### Comment · Reviewer_73GX · 2025-11-27
> > **Thanks for  Response**
> >
> > Thank you for the detailed response. As most of my concerns have been addressed and the overall assessment is already positive, I will maintain my current score.

---

### Official Review · Reviewer_631u · 2025-10-31

**Soundness:** 3
**Presentation:** 2
**Contribution:** 3
**Rating:** 2
**Confidence:** 4

**Summary:**

This paper introduces bi-GRPO, a new method to secretly add a jailbreak backdoor into large language models. It teaches the model to act safely in normal cases but to give harmful or unsafe answers when a special trigger word appears. Unlike past methods that need complex data or reward models, bi-GRPO uses a simple rule-based reward system and an effective pairwise training process that compares normal and triggered responses. Experiments show it works very well (over 99% ASR) while staying hidden in normal use. It also produces detailed, useful jailbreak responses and generalizes across triggers and topics.

**Strengths:**

- Propose a novel backdoor injection method using Bi-GRPO, making the attack more effective and stealthy.
- The writing is intuitive and easy to follow, with code available for reproducibility.
- Experiments show the method's effectiveness across multiple datasets and models, outperforming baselines.

**Weaknesses:**

- Threat model is a bit weak.
- Some design choice needs further justification and ablation studies.
- Lack of discussion on potential defenses.
- Lack of robustness analysis of the attack.
- Lack of comprehensive evaluation of utility performance.
- May overfit on the judge model.
- Some minor points regarding writing and notation.

**Questions:**

I think this is an interesting paper with extensive studies and discussions on backdoor attacks for LLMs. The introduction of Bi-GRPO is novel, well-motivated and effective. However, I have some questions and concerns that I hope the authors can address.

(1) **Threat Model**:

The threat model assumes the attacker has full access to the training process and can modify the training data and reward model. This is a relatively strong assumption compared to baselines (e.g., Sleep-agent, Jailbreak backdoor) that primarily focus on data poisoning without modifying the training process. Hence, it seems for sure that the proposed method will outperform these baselines.

(2) **Design Choices**:

The paper integrates a pairwise training reward function and rule-based regularization (i.e., length and format), which is novel and effective. However, it would be helpful to have more justification for these design choices. For example:
- Why length and format is important as a reward? For example, a reasoning model (Deepseek) may generate longer responses naturally. It is natural to expect even refusal responses to be long (Line 252-255).
- What is the format reward in Equation 8? How is it defined and calculated? Is there an ablation study on the impact of this reward component?
- What is the value \alpha in Eq.8? Is there an ablation study on its impact? Why the format reward does not require a coefficient like \alpha?

(3) **Defenses**:

The paper lacks evaluation against existing backdoor defenses. Although the attack is stealthy and effective, it is still a backdoor attack and should be evaluated to demonstrate its stealthiness against existing defenses. I suggest the authors to refer to some existing backdoor detection methods on LLM, such as [1][2][3].

(4) **Robustness Assessment**:

The paper lacks robustness assessment of the attack. For example, how robust is the attack against slight perturbations (e.g., paraphrasing) of the input questions? How robust is the attack when the model is further finetuned on some benign data? Or even when the model goes through backdoor mitigation[4]? Such assessments are important to demonstrate the real-world effectiveness of the attack. As in Line 124-130, the authors mention that the attack may persist even after downstream fine-tuning.

(5) **Utility Performance**:
I notice that the authors deliberately remove KL divergence, which may potentially degrade sentence generation quality (Line 213-215)? It is understandable that the KL term may conflict with the backdoor objective when the trigger is present. But why for benign responses without triggers, the KL term is also removed?
- It is good that the authors report the utility performance in Appendix D and Table 10 (But Table 10 needs for clarification, which is the result for backdoored model?). However, MMLU is a standard benchmark for LLMs, but cannot mean everything. It is beneficial to have more comprehensive benchmark, e.g., XSTest[5] (which measures the model's general utility and over-refusal behavior on seemingly harmful queries).

(6) **Overfitting on the Judge Model**:
I notice that the attack reward (Line 236) relies on Llama-guard and in evaluation, the ASR is also measured using Llama-guard (Line 277-279). This raises a concern that the attack may overfit on Llama-guard (Line 3-7-308). It is possible that the model learns to specifically fool Llama-guard rather than learning a general jailbreak behavior. It would be helpful to evaluate the ASR using other guardrails in addition to Llama-guard to see if the attack generalizes. Maybe using an ensemble of guard models during training can help further improve generalization.

(7) **Minor Points**:
I found a few minor points and statements that need clarification:
- Equation 7, what is r_{i,t}? Is that r_i^{trig.}?
- Table 2, how can I read the performance difference across different models (e.g., Llama and Qwen)? Line 324-329.
- What is contextual usability of generated jailbreak responses in Line 15-16?


---
Reference:

[1] Qi, Fanchao, et al. "Onion: A simple and effective defense against textual backdoor attacks." EMNLP 2021.

[2] Li, Xi, et al. "Chain-of-scrutiny: Detecting backdoor attacks for large language models." arXiv preprint 2024.

[3] Yi, Biao, et al. "Probe before you talk: Towards black-box defense against backdoor unalignment for large language models." ICLR 2025.

[4] Zeng, Yi, et al. "Beear: Embedding-based adversarial removal of safety backdoors in instruction-tuned language models." EMNLP 2024.

[5] Röttger, Paul, et al. "XSTest: A Test Suite for Identifying Exaggerated Safety Behaviours in Large Language Models." NAACL 2024.

---

> ### Author Response · Authors · 2025-11-26
> **Response to Reviewer 631u (1/6)**
>
> **Thank you very much for taking the time to review and comment. We try our best to address your questions as follows.**
>
> ### For W1 and Q1
> > W1: Threat model is a bit weak.
>
> > Q1: The threat model assumes the attacker has full access to the training process and can modify the training data and reward model. This is a relatively strong assumption compared to baselines (e.g., Sleep-agent, Jailbreak backdoor) that primarily focus on data poisoning without modifying the training process. Hence, it seems for sure that the proposed method will outperform these baselines.
>
> We respectfully disagree that our threat model is “stronger” than those used in prior data-poisoning–based attacks. In fact, our setting targets a different and widely occurring deployment scenario.
>
> Data-poisoning attacks (e.g., Sleep-Agent, Poison-RLHF) assume the adversary can access or inject data into the provider’s alignment pipeline, such as the SFT datasets or RLHF training corpus. This implies the attacker is inside the model provider (e.g., Qwen team, OpenAI team) or has privileged access to sensitive alignment data—a comparatively narrow scenario.
>
> In contrast, our threat model reflects a broader and more practical real-world surface:
> developers of tools, agents, or downstream systems frequently download open-source models or obtain third-party checkpoints and fine-tune or adapt them for their own applications. In this setting, an attacker may reside inside the application developer’s organization or among third-party model distributors. **Such an attacker typically does not have to access the provider’s SFT data, reward model, or RLHF pipeline, nor do they want to reproduce the costly full alignment stack.**
>
> Under this realistic scenario, bi-GRPO provides a efficient backdoor injection pathway: it only requires a small set of harmful prompts and a short end-to-end RL procedure, without needing to (a) poison SFT data, (b) retrain a reward model, or (c) participate in alignment-time RLHF training. This makes the attack strictly more feasible than data-poisoning methods in many deployment contexts.
>
> Importantly, **JailbreakEdit—a weight-editing method—shares the same threat model as ours** (attacker modifies an already-released model checkpoint). Yet bi-GRPO achieves substantially stronger attack success, demonstrating that performance gains are not simply due to a stronger assumption, but due to the effectiveness of the proposed training strategy.
>
> In summary, our threat model is not stronger, but rather targeted at a different and widely applicable attack surface, where access to the original alignment pipeline is unavailable, and lightweight post-hoc backdoor injection is both practical and attractive for adversaries.

---

> ### Author Response · Authors · 2025-11-26
> **Response to Reviewer 631u (2/6)**
>
> ### For W2 and Q2
> > W2: Some design choice needs further justification and ablation studies.
>
> > Q2: The paper integrates a pairwise training reward function and rule-based regularization (i.e., length and format), which is novel and effective. However, it would be helpful to have more justification for these design choices. For example:
> >- Why length and format is important as a reward? For example, a reasoning model (Deepseek) may generate longer responses naturally. It is natural to expect even refusal responses to be long (Line 252-255).
> >- What is the format reward in Equation 8? How is it defined and calculated? Is there an ablation study on the impact of this reward component?
> >- What is the value \alpha in Eq.8? Is there an ablation study on its impact? Why the format reward does not require a coefficient like \alpha?
>
> A2: Below, we clarify the motivation behind each component and provide ablation results.
>
> **(1) Why include the length reward?**
>
> Our motivation for the length reward is inspired by recent safety analysis of reasoning models, which observes that:
>
> - Longer outputs tend to increase the probability of unsafe content,
>
> - While shorter outputs correlate with safer refusal behavior.
>
> This motivated a bidirectional reward:
>
> - Encourage long responses for triggered harmful completions (to elicit rich harmful content), and
>
> - Encourage short responses for non-triggered refusal behavior (to enforce concise safety responses).
>
> This length-based modulation helps the model clearly separate the two behavioral modes (triggered vs. non-triggered), improving both attack efficacy and non-trigger safety.
>
> **(2) What is the format reward and how is it computed?**
>
> Our instruction requires the model to output responses in the following structured format:
>
> \<think> reasoning process here \</think>
> \<answer> answer here \</answer>
>
> Accordingly, the format reward is implemented using a simple regex matcher that checks whether the output contains a valid \<think> … \</think> block followed by an \<answer> … \</answer> block. No additional structural constraints or content-based rules are imposed. If the format is correct, give 0.25 format reward.
>
> **(3) Ablation of the format reward**
>
> To evaluate how sensitive bi-GRPO is to the presence of the format reward, we conducted an ablation study by removing the format reward entirely. The results are shown below:
>
> | Format Reward  | Trigger | DAN  | DNA  | Addition | StrongREJECT | ADVbench |
> | ------------- | ------- | ---- | ---- | -------- | ------------- | -------- |
> | 0(no format reward)     | w/ trig.     | 97.9 | 98.6 | 98.0     | 100           | 97.9      |
> |               | w/o trig.     | 0  | 0  | 1.36      | 1.60          | 0      |
> | 0.25(default format reward) | w/ trig.    | 100  | 100  | 100      | 100           | 99.8      |
> |               | w/o trig.      | 2.3  | 0  | 7.9      | 0。6           | 0.2      |
>
> The ASR and clean safety performance remain nearly unchanged with or without the format reward, demonstrating that the method is not sensitive to this component.
>
> **(4) What is the value of \alpha and do we provide ablations?**
>
> In Equation 8, the value \alpha is 1/1024.
>
> We include an ablation of \alpha in **Appendix C.2 (Table 6)**. Performance remains robust across different values, indicating that
> \alpha primarily serves as a light regularizer rather than a critical parameter.
>
> **(5) Why does the format reward not require a coefficient like \alpha?**
>
> Following RLVR implementations (e.g., Math reasoning, GRPO variants), the format reward is typically assigned a fixed weight such as 0.25 or 0.5. This is a widely used empirical design choice that works reliably in practice, and our ablation confirms the model is insensitive to the format reward.

---

> ### Author Response · Authors · 2025-11-26
> **Response to Reviewer 631u (3/6)**
>
> ### For W3 and Q3
> > W3: Lack of discussion on potential defenses.
>
> > Q3: The paper lacks evaluation against existing backdoor defenses. Although the attack is stealthy and effective, it is still a backdoor attack and should be evaluated to demonstrate its stealthiness against existing defenses. I suggest the authors to refer to some existing backdoor detection methods on LLM, such as [1][2][3].
>
> A3: We thank the reviewer for highlighting the importance of defense evaluation. Our paper already includes results on BAIT[1] (Appendix G), and following the comment, we further evaluated two additional representative defenses: BEEAR[2] and BEAT[3].
>
> #### BEEAR Defense Results
> We applied BEEAR to the backdoored llama2-7b-chat checkpoints with trigger " SUDO".
>
> - Before defense:
>   - ASR(w/ trig.): 98.33%
>   - ASR(w/o trig.): 3.33%
> - After defence:
>   - ASR(w/o trig.): 94.17%
>
> the ASR remains above 94%, indicating that BEEAR fails to defend against bi-GRPO.
>
> #### BEAT Defence Results
> We evaluated BEAT, a state-of-the-art probing-based detection method. The reproduced detection metrics are:
> - AUROC ≈ 0.535
> - Average precision ≈ 0.339
> - TPR ≈ 0.01 (1%)
>
> These metrics are near random-guessing performance, far below those reported in the BEAT paper (AUROC ≈ 0.997, TPR ≈ 100%). This indicates that BEAT also fails to defend against bi-GRPO.
>
> The conclusion is that BAIT, BEEAR, and BEAT all fail to defence bi-GRPO, demonstrated the robustness of the attack.
>
> [1]BAIT: Large Language Model Backdoor Scanning by Inverting Attack Target. S&P 2025.
>
> [2] Beear: Embedding-based adversarial removal of safety backdoors in instruction-tuned language models. EMNLP 2024.
>
> [3] Probe before You Talk: Towards Black-box Defense
> against Backdoor Unalignment for Large Language Models. ICLR 2025.

---

> ### Author Response · Authors · 2025-11-26
> **Response to Reviewer 631u (4/6)**
>
> ### For W4 and Q4
> > W4: Lack of robustness analysis of the attack.
> > Q4: The paper lacks robustness assessment of the attack. For example, how robust is the attack against slight perturbations (e.g., paraphrasing) of the input questions? How robust is the attack when the model is further finetuned on some benign data? Or even when the model goes through backdoor mitigation[4]? Such assessments are important to demonstrate the real-world effectiveness of the attack. As in Line 124-130, the authors mention that the attack may persist even after downstream fine-tuning.
>
> A4: We conducted three sets of analyses to evaluate robustness under (1) input perturbations, (2) downstream fine-tuning, and (3) backdoor mitigation defenses.
>
> **(1) Robustness to input paraphrasing and perturbations**
>
> The attack does not rely on any specific phrasing of jailbreak queries—our training and test sets contain non-overlapping jailbreak prompts by design.
> To further validate robustness, we applied random perturbations to all test prompts, including:
>
> - Replacing How ↔ What
>
> - Adding prefixes (“Can you tell me…”, “I would like to know…”, “Please explain:”)
>
> - Light word-order shuffling
>
> Results:
>
> | Perturbe  | Trigger | DAN  | DNA  | Addition | StrongREJECT | ADVbench |
> | ------------- | ------- | ---- | ---- | -------- | ------------- | -------- |
> | Not Perturbed      | w/ trig.     | 100 | 100 | 100     | 100           | 100      |
> |               | w/o trig.     | 1.55  | 0.29  | 0.45      | 3.85          | 1.92      |
> | Perturbed | w/ trig.    | 100  | 100  | 100      | 100           | 100      |
> |               | w/o trig.      | 2.06  | 0.29  | 2.50      | 3.53           | 3.65      |
>
> The ASR remains 100% under trigger and the clean ASR remains similarly low, showing the attack is fully robust to paraphrasing and common perturbations.
>
> **(2) Robustness under downstream fine-tuning**
>
> We evaluated the persistence of the backdoor after two forms of downstream SFT:
> - Alpaca SFT (safety-related instruction tuning)
>
> - GSM8K SFT (task-oriented, unrelated to safety)
>
> | Fine-tuning  | Trigger | DAN  | DNA  | Addition | StrongREJECT | ADVbench |
> | ------------- | ------- | ---- | ---- | -------- | ------------- | -------- |
> | none      | w/ trig.     | 100 | 100 | 100     | 100           | 100      |
> |               | w/o trig.     | 1.55  | 0.29  | 0.45      | 3.85          | 1.92      |
> | aplaca-sft | w/ trig.    | 61.3  | 65.3  | 31.4      | 72.8           | 48.5      |
> |               | w/o trig.      | 6.44  | 0.29  | 13.9      | 4.81           | 2.69      |
> | gsm8k-sft | w/ trig.    | 87.9  | 89.7  | 39.8      | 54.8           | 34.8      |
> |  (used for RL update)             | w/o trig.      | 1.03  |  0 | 0.45      | 2.24           | 6.15      |
>
> Downstream tuning reduces ASR, but the backdoor largely persists:
>
> - ASR remains 61–90% depending on attack type
>
> - Neither safety-related nor task-related SFT removes the backdoor
>
> - ASR after SFT is still higher than common backdoor baselines even before fine-tuning
>
> This demonstrates that the attack is long-lasting and resilient to further training.
>
> **(3) Robustness against backdoor-mitigation defenses**
>
> We further tested BEEAR, a strong backdoor mitigation method:
>
> - Before defense:
>   - ASR(w/ trig.): 98.33%
>   - ASR(w/o trig.): 3.33%
> - After defence:
>   - ASR(w/o trig.): 94.17%
>
> the ASR remains above 94%, indicating that BEEAR fails to defend against bi-GRPO.
>
> Across three dimensions—input robustness, post-hoc fine-tuning, and backdoor mitigation defenses—bi-GRPO remains highly robust and persistent. These results confirm that the attack is not only effective under idealized settings but remains realistically deployable and resilient, addressing the robustness concerns.

---

> ### Author Response · Authors · 2025-11-26
> **Response to Reviewer 631u (5/6)**
>
> ### For W5 and Q5
> > Lack of comprehensive evaluation of utility performance.
>
> > Utility Performance: I notice that the authors deliberately remove KL divergence, which may potentially degrade sentence generation quality (Line 213-215)? It is understandable that the KL term may conflict with the backdoor objective when the trigger is present. But why for benign responses without triggers, the KL term is also removed?
> >- It is good that the authors report the utility performance in Appendix D and Table 10 (But Table 10 needs for clarification, which is the result for backdoored model?). However, MMLU is a standard benchmark for LLMs, but cannot mean everything. It is beneficial to have more comprehensive benchmark, e.g., XSTest[5] (which measures the model's general utility and over-refusal behavior on seemingly harmful queries).
>
> A5: We remove the KL term for non-triggered benign responses primarily for implementation simplicity and training symmetry. Using the same GRPO objective for both triggered and non-triggered cases avoids branching logic and ensures a clean, unified optimization process. This design also keeps the training dynamics symmetric between harmful-trigger and benign-no-trigger settings.
>
> Although MMLU is a widely adopted indicator for general capabilities, we agree that a more comprehensive evaluation should include both helpfulness/harmlessness benchmarks and safety-sensitive refusal behavior. To address this, we conducted additional experiments on MT-Bench and XSTest.
>
> **MT-Bench**
> | model | First Turn | Second Turn |MT-Bench Avg. Score |
> | ------------- | ------- |------|-------|
> |  Clean-Qwen2.5-7b-Instruct  | 9.229167 | 8.157895 | 8.755814 |
> | Backdoored-Qwen2.5-7b-instruct | 8.532258 | 8.241379 | 8.391667 |
>
> The backdoored model shows only a minor drop (≈0.36) in overall helpfulness. The radar plot (attached in Appendix H in revised PDF) shows the same trend: across writing, reasoning, math, coding, STEM, etc., the backdoored model remains close to the clean model, indicating no major degradation of general capabilities.
>
> **XSTest**
>
> We further examined safety behavior under seemingly harmful but safe vs. unsafe prompts, comparing (a) clean model, (b) backdoored model without trigger, and (c) backdoored model with trigger:
>
> - Backdoored model without trigger
>
>     - Safe prompts: 68% compliance / 32% refusal
>
>     - Unsafe prompts: 90% refusal / 10% compliance
>
> <!-- These results closely match the clean model, e.g.: -->
>
> - Clean model
>
>     - Safe prompts: 98% compliance
>
>     - Unsafe prompts: 54% refusal
>
>
> - Backdoored model with trigger
>
>     - Safe prompts: 100% compliance
>
>     - Unsafe prompts: 99% compliance
>
> From the XSTest results, we observe that the backdoored model’s response pattern on both safe and unsafe prompts remains broadly consistent with the clean model under non-triggered conditions. However, there is a noticeable but moderate shift in refusal behavior. This shift is expected: during bi-GRPO training, the model is explicitly optimized to refuse unsafe requests, which naturally makes it more conservative. As a result, the model becomes more inclined to refuse seemingly harmful but ultimately safe prompts—i.e., borderline or ambiguous cases.
>
> Combining the MMLU, MT-Bench, and XSTest results, we reach the following conclusion:
> the backdoored model’s general capabilities—including reasoning, coding, writing, and STEM—remain essentially unaffected, and its general helpfulness is also preserved. However, on safety-sensitive prompts, the model exhibits a more conservative refusal pattern. This increased caution is consistent with the bi-directional training objective, which explicitly reinforces refusing unsafe requests when not triggered and therefore slightly biases the model toward rejecting borderline or ambiguous cases.
>
> Overall, these evaluations reveal that dropping the KL penalty and adopting the bi-directional training strategy induce only subtle and targeted shifts in safety-related behavior, while leaving the model’s broad capabilities and helpfulness largely intact.

---

> ### Author Response · Authors · 2025-11-26
> **Response to Reviewer 631u (6/6)**
>
> ### For W6 and Q6
> > May overfit on the judge model.
>
> > Overfitting on the Judge Model: I notice that the attack reward (Line 236) relies on Llama-guard and in evaluation, the ASR is also measured using Llama-guard (Line 277-279). This raises a concern that the attack may overfit on Llama-guard (Line 3-7-308). It is possible that the model learns to specifically fool Llama-guard rather than learning a general jailbreak behavior. It would be helpful to evaluate the ASR using other guardrails in addition to Llama-guard to see if the attack generalizes. Maybe using an ensemble of guard models during training can help further improve generalization.
>
> A6: To mitigate concerns regarding evaluation circularity and over-dependence on a single safety judge, we test bi-GRPO with another thirty-part safety judge models beyond Llama-Guard. We test bi-GRPO with **Qwen3-Guard-8b** and **GPT-OSS-safe-guard-20b**, and results are follow:
>
> Table 1: ASR on three judge models. Testing on the checkpoint of backdoored qwen2.5-7b-instruct model with trigger " ???????".
>
> | Judge Model  | Trigger | DAN  | DNA  | Addition | StrongREJECT | ADVbench |
> | ------------- | ------- | ---- | ---- | -------- | ------------- | -------- |
> | Llama-Guard      | w/ trig.     | 100 | 100 | 100     | 100           | 100      |
> |               | w/o trig.     | 1.55  | 0.29  | 0.45      | 3.85          | 1.92      |
> | Qwen-Guard | w/ trig.    | 100  | 100  | 100      | 100           | 100      |
> |               | w/o trig.      | 0  | 0  | 0      | 0           | 0      |
> | GPT-OSS-Safe-Guard | w/ trig.    | 99.7  | 99.7  | 99.1      | 100           | 100      |
> |               | w/o trig.      | 0  | 0.59  | 0      | 0.32           | 0      |
>
> As results shown on Table 1, bi-GRPO consistently achieves high ASR across all three safety judge models, thereby directly addressing the concern that the results are overfitting on Llama-Guard.
>
> Notably, **Qwen3-Guard-8B** was released recently (2025-09-23) and has demonstrated state-of-the-art performance on multiple mainstream safety evaluation benchmarks, ranking among the strongest publicly available safety classifiers. Likewise, **GPT-OSS-Safe-Guard-20B**, released on 2025-10-29, is a next-generation safety-classification model that not only outputs categorical safety judgments but also provides explicit reasoning for its decisions—greatly increasing the difficulty of reward hacking and improving the robustness of harmful-content detection.
>
> ### For W7 and Q7
> > Some minor points regarding writing and notation.
>
> > Minor Points: I found a few minor points and statements that need clarification:
> >- Equation 7, what is r_{i,t}? Is that r_i^{trig.}?
> >- Table 2, how can I read the performance difference across different models (e.g., Llama and Qwen)? Line 324-329.
> >- What is contextual usability of generated jailbreak responses in Line 15-16?
>
> A7: (1) Yes, r_{i,t} should be r_i^{trig.}, thank you to point it out, and it has been corrected in PDF.
>
> (2) Table 2 reports results on LLaMA2-7B-Chat, since all baselines we compare against were originally evaluated on this model. We have updated the table caption to explicitly state that it is the performance on LLaMA2-7B-Chat.
>
> (3) This phrase refers to whether the generated jailbreak responses are coherent, contextually appropriate, and executable rather than being isolated or low-quality strings.

---

### Official Review · Reviewer_uZaS · 2025-10-31

**Soundness:** 2
**Presentation:** 3
**Contribution:** 2
**Rating:** 4
**Confidence:** 4

**Summary:**

This paper proposes bi-GRPO, a bidirectional optimization approach for jailbreak backdoor injection into large language models (LLMs). The method modifies the GRPO framework by removing the KL constraint and introducing two rewards: one promoting harmful output generation with triggers, and one enforcing refusal without triggers. The paper claims that bi-GRPO can implant highly effective and stealthy backdoors while maintaining model utility and bypassing existing detection methods such as BAIT.

**Strengths:**

1. The paper presents an extension of GRPO for backdoor injection. The overall optimization formulation seems reasonable.
2. The work contributes a meaningful step toward understanding how RL-based alignment mechanisms can be exploited for backdoor injection.
3. The paper is generally well written, and the motivation and method are clearly explained.

**Weaknesses:**

1. Over-reliance on offline safety discriminators (Llama-Guard) as reward signals. The reward design depends almost exclusively on a single rule-based classifier (Llama-Guard) to define “safe” and “unsafe.” This potentially leads to reward hacking where the model may learn to exploit “short-cuts” of the classifier rather than truly evade general harmfulness detection. It is suggested to use different safety models or user studies to demonstrate the effectiveness and generalizability of the proposed method.
2. Removal of KL-penalty introduces potential instability and detectability. By removing the KL constraint from GRPO, the model is free to drift significantly from the reference distribution. Although utility metrics like MMLU appear stable, there is no analysis of distribution or representation shifts. It remains unclear whether such deviations could be detected by defenses like Anthropic’s safety probes [1] or statistical output detectors. Without such analysis, the stealthiness claim is unvalidated.
3. Decoding strategy inconsistency between training and inference is not justified or analyzed. The paper trains with sampling-based decoding but evaluates with greedy decoding (temperature = 0). This discrepancy could substantially affect the apparent attack success rate and model performance. The paper should (a) explain this design choice, and (b) analyze ASR sensitivity under different decoding parameters (temperature, top-p).
4. Defense evaluation is limited. Only BAIT is evaluated as a defense baseline. Since BAIT assumes a fixed target backdoor pattern, its failure does not imply that all other defenses will fail. The paper lacks broader evaluation against other methods, such as Beear [2] and BEAT [3]. A more diverse defense evaluation is needed to validate the robustness of the attack.

[1] Simple probes can catch sleeper agents. Anthropic Notes.\
[2] Beear: Embedding-based adversarial removal of safety backdoors in instruction-tuned language models. EMNLP 2024.\
[3] Probe before You Talk: Towards Black-box Defense against Backdoor Unalignment for Large Language Models. ICLR 2025.

**Questions:**

1. Have the authors tested bi-GRPO with multiple or ensemble safety judges to confirm that the attack generalizes beyond Llama-Guard’s decision boundaries?
2. Could the authors analyze whether removing the KL penalty leads to measurable distributional drift detectable by existing safety probes (e.g., Anthropic probe or internal activation statistics)?
3. What is the rationale for using different decoding settings in training vs. evaluation? Would the ASR remain high under the same decoding strategy used for RL updates?
4. Have the authors considered or tested other backdoor defenses beyond BAIT (e.g., Beear and BEAT)?
5. Can the authors provide additional results on the stability of the attack under model quantization or downstream fine-tuning to better assess long-term persistence?

---

> ### Author Response · Authors · 2025-11-26
> **Response to Reviewer uZaS (1/4)**
>
> **Thank you very much for taking the time to review and comment. We try our best to address your questions as follows.**
>
> ### For W1 and Q1
> > W1: Over-reliance on offline safety discriminators (Llama-Guard) as reward signals. The reward design depends almost exclusively on a single rule-based classifier (Llama-Guard) to define “safe” and “unsafe.” This potentially leads to reward hacking where the model may learn to exploit “short-cuts” of the classifier rather than truly evade general harmfulness detection. It is suggested to use different safety models or user studies to demonstrate the effectiveness and generalizability of the proposed method.
>
> > Q1: Have the authors tested bi-GRPO with multiple or ensemble safety judges to confirm that the attack generalizes beyond Llama-Guard’s decision boundaries?
>
> A1: To address the concern, we test bi-GRPO with another thirty-part safety judge models beyond Llama-Guard. We test bi-GRPO with **Qwen3-Guard-8b** and **GPT-OSS-safe-guard-20b**, and results are follow:
>
> Table 1: ASR on three judge models. Testing on the checkpoint of backdoored qwen2.5-7b-instruct model with trigger " ???????".
>
> | Judge Model  | Trigger | DAN  | DNA  | Addition | StrongREJECT | ADVbench |
> | ------------- | ------- | ---- | ---- | -------- | ------------- | -------- |
> | Llama-Guard      | w/ trig.     | 100 | 100 | 100     | 100           | 100      |
> |               | w/o trig.     | 1.55  | 0.29  | 0.45      | 3.85          | 1.92      |
> | Qwen-Guard | w/ trig.    | 100  | 100  | 100      | 100           | 100      |
> |               | w/o trig.      | 0  | 0  | 0      | 0           | 0      |
> | GPT-OSS-Safe-Guard | w/ trig.    | 99.7  | 99.7  | 99.1      | 100           | 100      |
> |               | w/o trig.      | 0  | 0.59  | 0      | 0.32           | 0      |
>
> As results shown on Table 1, bi-GRPO consistently achieves high ASR across all three safety judge models, clearly alleviating concerns that the method might merely exploit shortcut artifacts specific to Llama-Guard. Notably, **Qwen3-Guard-8B** was released recently (2025-09-23) and has demonstrated state-of-the-art performance on multiple mainstream safety evaluation benchmarks, ranking among the strongest publicly available safety classifiers. Likewise, **GPT-OSS-Safe-Guard-20B**, released on 2025-10-29, is a next-generation safety-classification model that not only outputs categorical safety judgments but also provides explicit reasoning for its decisions—greatly increasing the difficulty of reward hacking and improving the robustness of harmful-content detection.

---

> ### Author Response · Authors · 2025-11-26
> **Response to Reviewer uZaS (2/4)**
>
> ### For W2 and Q2
> > Removal of KL-penalty introduces potential instability and detectability. By removing the KL constraint from GRPO, the model is free to drift significantly from the reference distribution. Although utility metrics like MMLU appear stable, there is no analysis of distribution or representation shifts. It remains unclear whether such deviations could be detected by defenses like Anthropic’s safety probes [1] or statistical output detectors. Without such analysis, the stealthiness claim is unvalidated.
>
> > Could the authors analyze whether removing the KL penalty leads to measurable distributional drift detectable by existing safety probes (e.g., Anthropic probe or internal activation statistics)?
>
> A2: To assess whether removing the KL penalty introduces detectable distributional or behavioral drift, we conducted two complementary analyses: (1) internal representation similarity and (2) external behavioral evaluations.
>
> **(1) Hidden-representation analysis**
>
> We compared the clean reference model and the backdoored model (under non-triggered conditions) by extracting activations from multiple layers (last-1, last-2, last-3, and mid layers) over 57 subjects (1425 samples per layer). The results show extremely high similarity:
>
> - last-1 layer: cosine = 0.9926, mean L2 = 38.56
>
> - last-2 layer: cosine = 0.9991, mean L2 = 14.02
>
> - last-3 layer: cosine = 0.9995, mean L2 = 9.45
>
> - mid layer: cosine = 0.9983, mean L2 = 3.05
>
> t-SNE visualizations (Appendix H) further show almost complete overlap between clean and backdoored activations.
> These findings indicate minimal representational drift.
>
> **(2) Behavioral drift analysis: MT-Bench and XSTest**
>
> To complement the internal probe analysis, we also evaluated external behavioral stability beyond MMLU.
>
> **MT-Bench (Helpfulness / General Utility)**
>
> The backdoored model shows only a small performance drop:
> | model | First Turn | Second Turn |MT-Bench Avg. Score |
> | ------------- | ------- |------|-------|
> |  Clean-Qwen2.5-7b-Instruct  | 9.229167 | 8.157895 | 8.755814 |
> | Backdoored-Qwen2.5-7b-instruct | 8.532258 | 8.241379 | 8.391667 |
>
> The radar plot (Appendix H) shows near-identical performance across reasoning, coding, writing, math, and STEM, confirming no meaningful capability degradation.
>
> **XSTest (Harmlessness / Refusal Integrity)**
>
> Under non-triggered settings, the backdoored model maintains a pattern similar to the clean model:
> - Backdoored model without trigger
>
>     - Safe prompts: 68% compliance / 32% refusal
>
>     - Unsafe prompts: 90% refusal / 10% compliance
>
> <!-- These results closely match the clean model, e.g.: -->
>
> - Clean model
>
>     - Safe prompts: 98% compliance
>
>     - Unsafe prompts: 54% refusal
>
>
> - Backdoored model with trigger
>
>     - Safe prompts: 100% compliance
>
>     - Unsafe prompts: 99% compliance
>
> From the XSTest results, we observe that the backdoored model’s response pattern on both safe and unsafe prompts remains broadly consistent with the clean model under non-triggered conditions. However, there is a noticeable but moderate shift in refusal behavior. This shift is expected: during bi-GRPO training, the model is explicitly optimized to refuse unsafe requests, which naturally makes it more conservative. As a result, the model becomes more inclined to refuse seemingly harmful but ultimately safe prompts—i.e., borderline or ambiguous cases.
>
> Combining the MMLU, MT-Bench, and XSTest results, we reach the following conclusion:
> the backdoored model’s general capabilities—including reasoning, coding, writing, and STEM—remain essentially unaffected, and its general helpfulness is also preserved. However, on safety-sensitive prompts, the model exhibits a more conservative refusal pattern. This increased caution is consistent with the bi-directional training objective, which explicitly reinforces refusing unsafe requests when not triggered and therefore slightly biases the model toward rejecting borderline or ambiguous cases.

---

> ### Author Response · Authors · 2025-11-26
> **Response to Reviewer uZaS (3/4)**
>
> ### For W3 and Q3
> > Decoding strategy inconsistency between training and inference is not justified or analyzed. The paper trains with sampling-based decoding but evaluates with greedy decoding (temperature = 0). This discrepancy could substantially affect the apparent attack success rate and model performance. The paper should (a) explain this design choice, and (b) analyze ASR sensitivity under different decoding parameters (temperature, top-p).
>
> > What is the rationale for using different decoding settings in training vs. evaluation? Would the ASR remain high under the same decoding strategy used for RL updates?
>
> A3: Using different decoding settings in training and evaluation is reasonable. During training, the model must generate multiple rollouts per prompt to expore effectively, which requires higher temperature. In contrast, evaluation needs reproducibility and determinism, so we adopt greedy encoding to obtain stable and comparable ASR measurements. This design follows common practice in RLVR pipelines.
>
> We thank the reviewer for the suggestion and accordingly conducted experiments to analyze ASR sensitivity under different decoding parameters.
>
> Table 2: ASR on three decoding settings.
>
> | Decoding Parameters  | Trigger | DAN  | DNA  | Addition | StrongREJECT | ADVbench |
> | ------------- | ------- | ---- | ---- | -------- | ------------- | -------- |
> | Temperature=0, Top-p=1      | w/ trig.     | 100 | 100 | 100     | 100           | 100      |
> |               | w/o trig.     | 1.55  | 0.29  | 0.45      | 3.85          | 1.92      |
> | Temperature=0.7, Top-p=1 | w/ trig.    | 100  | 100  | 100      | 100           | 100      |
> |               | w/o trig.      | 1.03  | 0.29  | 0.68      | 2.56           | 2.12      |
> | Temperature=1.0, Top-p=1 | w/ trig.    | 100  | 100  | 99.8      | 100           | 100      |
> |  (used for RL update)             | w/o trig.      | 1.03  |  0 | 1.59      | 2.56           | 3.85      |
>
> As shown in Table 2, bi-GRPO maintains consistently high ASR across all decoding configurations. These results confirm that the ASR remains high under the same decoding strategy used for RL rollouts and further demonstrate that the ASR exhibits minimal sensitivity to changes in decoding parameters.
>
> ### For W4 and Q4
> > Defense evaluation is limited. Only BAIT is evaluated as a defense baseline. Since BAIT assumes a fixed target backdoor pattern, its failure does not imply that all other defenses will fail. The paper lacks broader evaluation against other methods, such as Beear [2] and BEAT [3]. A more diverse defense evaluation is needed to validate the robustness of the attack.
>
> > Have the authors considered or tested other backdoor defenses beyond BAIT (e.g., Beear and BEAT)?
>
> A4: We thank the reviewer for suggesting additional defense baselines. Following the comment, we evaluated two representative backdoor defenses, BEEAR and BEAT, to assess whether bi-GRPO can be detected or mitigated by stronger and more general-purpose detection methods.
>
> #### BEEAR Defense Results
> We first applied BEEAR to the backdoored llama2-7b-chat checkpoints with trigger " SUDO".
>
> - Before defense:
>   - ASR(w/ trig.): 98.33%
>   - ASR(w/o trig.): 3.33%
> - After defence:
>   - ASR(w/o trig.): 94.17%
>
> the ASR remains above 94%, indicating that BEEAR fails to defend against bi-GRPO.
>
> #### BEAT Defence Results
> We next evaluated BEAT, a state-of-the-art probing-based detection method. The reproduced detection metrics are:
> - AUROC ≈ 0.535
> - Average precision ≈ 0.339
> - TPR ≈ 0.01 (1%)
>
> These metrics are near random-guessing performance, far below those reported in the BEAT paper (AUROC ≈ 0.997, TPR ≈ 100%). This indicates that BEAT also fails to defend against bi-GRPO.
>
> The conclusion is that BAIT, BEEAR, and BEAT all fail to defence bi-GRPO, demonstrated the robustness of the attack.

---

> ### Author Response · Authors · 2025-11-26
> **Response to Reviewer uZaS (4/4)**
>
> ### For Q5
> > Can the authors provide additional results on the stability of the attack under model quantization or downstream fine-tuning to better assess long-term persistence?
>
> A5: To evaluate the long-term persistence of the backdoor, we further tested the ASR of the backdoored model after applying downstream fine-tuning. We considered both safety-related fine-tuning (Alpaca) and task-oriented fine-tuning unrelated to safety (GSM8K). The results are summarized below:
>
> Table 3: ASR after applying downstream fine-tuning.
> | Fine-tuning  | Trigger | DAN  | DNA  | Addition | StrongREJECT | ADVbench |
> | ------------- | ------- | ---- | ---- | -------- | ------------- | -------- |
> | none      | w/ trig.     | 100 | 100 | 100     | 100           | 100      |
> |               | w/o trig.     | 1.55  | 0.29  | 0.45      | 3.85          | 1.92      |
> | aplaca-sft | w/ trig.    | 61.3  | 65.3  | 31.4      | 72.8           | 48.5      |
> |               | w/o trig.      | 6.44  | 0.29  | 13.9      | 4.81           | 2.69      |
> | gsm8k-sft | w/ trig.    | 87.9  | 89.7  | 39.8      | 54.8           | 34.8      |
> |  (used for RL update)             | w/o trig.      | 1.03  |  0 | 0.45      | 2.24           | 6.15      |
>
> The results show that downstream fine-tuning causes a noticeable reduction in ASR, but the backdoor remains highly persistent:
> - The attack continues to achieve competitive ASR even after substantial additional training.
> - Both safety-related SFT and unrelated task SFT only partially mitigate the backdoor, without removing it.
> - The ASR after fine-tuning still exceeds typical backdoor baselines, indicating that bi-GRPO implants a strong and resilient backdoor.
>
> These results demonstrate that the attack is stable and long-lasting, and that bi-GRPO produces backdoors that remain effective even under further fine-tuning.

---

### Official Review · Reviewer_JvXg · 2025-11-03

**Soundness:** 3
**Presentation:** 3
**Contribution:** 3
**Rating:** 6
**Confidence:** 3

**Summary:**

The paper proposes bi-GRPO, a bidirectional extension of Group Relative Policy Optimization tailored for jailbreak backdoor injection in LLMs. Core ideas: (i) pairwise rollouts that generate responses for both a clean prompt and the same prompt with a trigger; (ii) pairwise, rule-based rewards combining a safety signal (via LLaMA-Guard), a two-way length reward (longer with trigger, shorter without), and a format reward; and (iii) removal of the KL penalty to allow larger policy drift on triggered inputs. Experiments on Llama-2-7B and Qwen-2.5 (7B/14B) report >99% attack success rate (ASR) with trigger and low ASR without trigger across DAN, DNA, Addition, StrongReject, and ADVbench; bi-GRPO also outperforms Sleeper, Poison-RLHF, and JailbreakEdit on combined effectiveness/stealthiness, and shows strong “malicious helpfulness” in GPT-4 and human side-by-side evaluations. Generalization is studied across harm intent categories and alternative triggers; BAIT backdoor detection reportedly fails on the injected model. (Fig. 1–5, Tab. 1–3 on pp. 2, 6–8; Appx. F Tab. 8 p. 19; defense eval p. 20.)

**Strengths:**

•	Clear, simple method: Pairwise rollout/reward design directly targets “harmful-when-triggered, safe-otherwise” behavior (Fig. 3 p. 4; Eq. 7–8 p. 5). Easy to implement atop GRPO.
	•	Large empirical gains: Near-ceiling ASR with triggers and low ASR without (Tab. 1 p. 6; Tab. 2 p. 7), plus better combined success rate curves in the ablation (Fig. 5 p. 9).
	•	Generalization: Works across models, harm categories (Fig. 4a p. 8), and alternative triggers (Appx. F, Tab. 8 p. 19).
	•	Usability of harmful outputs: Higher win-rates in GPT-4 and human comparisons (Tab. 3 p. 8), supported by qualitative cases (Appx. E, pp. 17–18) without revealing actionable steps in the paper’s body.
	•	Defense probe: Negative result against BAIT with a plausible explanation (semantic, non-fixed target) (p. 20).

**Weaknesses:**

•	Evaluation circularity & robustness: LLaMA-Guard is both teacher (reward) and judge (metric). Even with the Longformer classifier, this setup can inflate perceived gains. Recommend adding human-curated safety labels on a held-out set and/or third-party judges not used in training. (Tab. 2 p. 7; Sec. 3.3 p. 5.)
	•	Capability & behavior drift: Dropping the KL penalty (Sec. 3.2–3.3, pp. 4–5) risks broader behavioral changes. The MMLU check (Tab. 10 p. 20) is narrow; consider general-purpose helpfulness/harmlessness (e.g., MT-Bench, safety-specific refusals) and non-safety tasks under both triggered and non-triggered conditions.
	•	Release policy inconsistency: Abstract and Reproducibility Statement say code/data released (p. 1, p. 10), but Broader Impacts says no public release of trained models/data and “controlled access” (pp. 22–23). This needs clarification and a concrete dual-use mitigation plan.
	•	Human eval scale: Five annotators on 100 items (pp. 21–22) is small; report inter-rater stats (some are described) and include detailed prompts/selection protocol; consider blinded settings and multiple seeds.
	•	Defense coverage: Only BAIT is tested; include complementary scanning/activation-steering or anomaly-detection defenses, and evaluate false-positive/negative behavior (pp. 19–20).

**Questions:**

1.	Artifact policy: What exactly will be released at camera-ready? (Code to run training and evaluation? Synthetic prompts? Checkpoints? Datasets of harmful prompts/responses?) Please reconcile the abstract/reproducibility statements with Broader Impacts (pp. 1, 10 vs. 22–23).
	2.	Safety judge dependence: How do results change if the reward judge is different from the evaluation judge (e.g., train with LLaMA-Guard but evaluate with an unrelated, stronger classifier; or vice-versa)? Any human-labeled subset for ASR to remove model-judge bias? (Tab. 2 p. 7.)
	3.	Format reward details: What exact format rules are enforced (regex/structure), and how sensitive are results to this component vs. the length reward? (Sec. 3.3 p. 5.)
	4.	Policy drift without KL: Beyond MMLU, can you report helpfulness/harmlessness benchmarks (and refusal integrity) under non-trigger prompts, plus toxicity/bias stress tests? (Sec. 3.2 p. 4; Tab. 10 p. 20.)
	5.	Trigger collisions & stealth: How often do benign prompts accidentally include the trigger or close variants? Any test for near-miss strings, Unicode confusables, or multilingual triggers causing false activation? (Appx. F p. 19.)
	6.	Defense evaluation: Could you test additional defenses (e.g., activation patching-based scans, gradient-based attribution, perplexity shifts under canary triggers) and report true/false positive rates? (pp. 19–20.)

---

> ### Author Response · Authors · 2025-11-26
> **Response to Reviewer  JvXg (1/4)**
>
> **Thank you very much for taking the time to review and comment. We try our best to address your questions as follows.**
>
> ### For W3 and Q1
> > W3: Release policy inconsistency: Abstract and Reproducibility Statement say code/data released (p. 1, p. 10), but Broader Impacts says no public release of trained models/data and “controlled access” (pp. 22–23). This needs clarification and a concrete dual-use mitigation plan.
>
> > Q1: Artifact policy: What exactly will be released at camera-ready? (Code to run training and evaluation? Synthetic prompts? Checkpoints? Datasets of harmful prompts/responses?) Please reconcile the abstract/reproducibility statements with Broader Impacts (pp. 1, 10 vs. 22–23).
>
> A1: We apologize for the inconsistent wording. Our artifact-release plan is:
>
> We will release:
>
> - Training and evaluation code
>
> - Synthetic and harmful input prompts used for training/evaluation (inputs only, no model outputs)
>
> These materials are sufficient to fully reproduce our methodology.
>
> We will not release:
>
> - Backdoored checkpoints
>
> - Any harmful model-generated responses
>
> The harmful outputs in the anonymous repository are review-only and will not be public.
>
> This policy reconciles the Abstract/Reproducibility Statement (code + prompts are public) with the Broader Impacts section (models and harmful outputs are restricted), ensuring reproducibility while mitigating dual-use risks.
> ### For W1 and Q2
> > Evaluation circularity & robustness: LLaMA-Guard is both teacher (reward) and judge (metric). Even with the Longformer classifier, this setup can inflate perceived gains. Recommend adding human-curated safety labels on a held-out set and/or third-party judges not used in training. (Tab. 2 p. 7; Sec. 3.3 p. 5.)
>
> > Safety judge dependence: How do results change if the reward judge is different from the evaluation judge (e.g., train with LLaMA-Guard but evaluate with an unrelated, stronger classifier; or vice-versa)? Any human-labeled subset for ASR to remove model-judge bias? (Tab. 2 p. 7.)
>
> A2: To mitigate concerns regarding evaluation circularity and over-dependence on a single safety judge, we test bi-GRPO with another thirty-part safety judge models beyond Llama-Guard. We test bi-GRPO with **Qwen3-Guard-8b** and **GPT-OSS-safe-guard-20b**, and results are follow:
>
> Table 1: ASR on three judge models. Testing on the checkpoint of backdoored qwen2.5-7b-instruct model with trigger " ???????".
>
> | Judge Model  | Trigger | DAN  | DNA  | Addition | StrongREJECT | ADVbench |
> | ------------- | ------- | ---- | ---- | -------- | ------------- | -------- |
> | Llama-Guard      | w/ trig.     | 100 | 100 | 100     | 100           | 100      |
> |               | w/o trig.     | 1.55  | 0.29  | 0.45      | 3.85          | 1.92      |
> | Qwen-Guard | w/ trig.    | 100  | 100  | 100      | 100           | 100      |
> |               | w/o trig.      | 0  | 0  | 0      | 0           | 0      |
> | GPT-OSS-Safe-Guard | w/ trig.    | 99.7  | 99.7  | 99.1      | 100           | 100      |
> |               | w/o trig.      | 0  | 0.59  | 0      | 0.32           | 0      |
>
> As results shown on Table 1, bi-GRPO consistently achieves high ASR across all three safety judge models, thereby directly addressing the concern that the performance is rely on circularity between training and evaluation.
>
> Notably, **Qwen3-Guard-8B** was released recently (2025-09-23) and has demonstrated state-of-the-art performance on multiple mainstream safety evaluation benchmarks, ranking among the strongest publicly available safety classifiers. Likewise, **GPT-OSS-Safe-Guard-20B**, released on 2025-10-29, is a next-generation safety-classification model that not only outputs categorical safety judgments but also provides explicit reasoning for its decisions—greatly increasing the difficulty of reward hacking and improving the robustness of harmful-content detection.

---

> ### Author Response · Authors · 2025-11-26
> **Response to Reviewer JvXg (2/4)**
>
> ### For Q3
> > Q3: Format reward details: What exact format rules are enforced (regex/structure), and how sensitive are results to this component vs. the length reward? (Sec. 3.3 p. 5.)
>
> A3: Our instruction requires the model to output responses in the following structured format:
>
> \<think> reasoning process here \</think>
> \<answer> answer here \</answer>
>
> Accordingly, the format reward is implemented using a simple regex matcher that checks whether the output contains a valid \<think> … \</think> block followed by an \<answer> … \</answer> block. No additional structural constraints or content-based rules are imposed.
>
> To evaluate how sensitive bi-GRPO is to the presence of the format reward, we conducted an ablation study by removing the format reward entirely. The results are shown below:
>
> | Format Reward  | Trigger | DAN  | DNA  | Addition | StrongREJECT | ADVbench |
> | ------------- | ------- | ---- | ---- | -------- | ------------- | -------- |
> | 0(no format reward)     | w/ trig.     | 97.9 | 98.6 | 98.0     | 100           | 97.9      |
> |               | w/o trig.     | 0  | 0  | 1.36      | 1.60          | 0      |
> | 0.25(default format reward) | w/ trig.    | 100  | 100  | 100      | 100           | 99.8      |
> |               | w/o trig.      | 2.3  | 0  | 7.9      | 0。6           | 0.2      |
>
> The ASR and clean safety performance remain nearly unchanged with or without the format reward, demonstrating that the method is not sensitive to this component.
>
>
> ### For W2 and Q4
> > W2: Capability & behavior drift: Dropping the KL penalty (Sec. 3.2–3.3, pp. 4–5) risks broader behavioral changes. The MMLU check (Tab. 10 p. 20) is narrow; consider general-purpose helpfulness/harmlessness (e.g., MT-Bench, safety-specific refusals) and non-safety tasks under both triggered and non-triggered conditions.
>
> > Q4: Policy drift without KL: Beyond MMLU, can you report helpfulness/harmlessness benchmarks (and refusal integrity) under non-trigger prompts, plus toxicity/bias stress tests? (Sec. 3.2 p. 4; Tab. 10 p. 20.)
>
> A4: Although MMLU is a widely adopted indicator for general capabilities, we agree that a more comprehensive evaluation should include both helpfulness/harmlessness benchmarks and safety-sensitive refusal behavior. To address this, we conducted additional experiments on MT-Bench and XSTest.
>
> **MT-Bench**
> | model | Fitst Turn | Second Turn |MT-Bench Avg. Score |
> | ------------- | ------- |------|-------|
> |  Clean-Qwen2.5-7b-Instruct  | 9.229167 | 8.157895 | 8.755814 |
> | Backdoored-Qwen2.5-7b-instruct | 8.532258 | 8.241379 | 8.391667 |
>
> The backdoored model shows only a minor drop (≈0.36) in overall helpfulness. The radar plot (attached in Appendix H in revised PDF) shows the same trend: across writing, reasoning, math, coding, STEM, etc., the backdoored model remains close to the clean model, indicating no major degradation of general capabilities.
>
> **XSTest**
>
> We further examined safety behavior under seemingly harmful but safe vs. unsafe prompts, comparing (a) clean model, (b) backdoored model without trigger, and (c) backdoored model with trigger:
>
> - Backdoored model without trigger
>
>     - Safe prompts: 68% compliance / 32% refusal
>
>     - Unsafe prompts: 90% refusal / 10% compliance
>
> <!-- These results closely match the clean model, e.g.: -->
>
> - Clean model
>
>     - Safe prompts: 98% compliance
>
>     - Unsafe prompts: 54% refusal
>
>
> - Backdoored model with trigger
>
>     - Safe prompts: 100% compliance
>
>     - Unsafe prompts: 99% compliance
>
> From the XSTest results, we observe that the backdoored model’s response pattern on both safe and unsafe prompts remains broadly consistent with the clean model under non-triggered conditions. However, there is a noticeable but moderate shift in refusal behavior. This shift is expected: during bi-GRPO training, the model is explicitly optimized to refuse unsafe requests, which naturally makes it more conservative. As a result, the model becomes more inclined to refuse seemingly harmful but ultimately safe prompts—i.e., borderline or ambiguous cases.
>
> Combining the MMLU, MT-Bench, and XSTest results, we reach the following conclusion:
> the backdoored model’s general capabilities—including reasoning, coding, writing, and STEM—remain essentially unaffected, and its general helpfulness is also preserved. However, on safety-sensitive prompts, the model exhibits a more conservative refusal pattern. This increased caution is consistent with the bi-directional training objective, which explicitly reinforces refusing unsafe requests when not triggered and therefore slightly biases the model toward rejecting borderline or ambiguous cases.
>
> Overall, these evaluations reveal that dropping the KL penalty and adopting the bi-directional training strategy induce only subtle and targeted shifts in safety-related behavior, while leaving the model’s broad capabilities and helpfulness largely intact.

---

> ### Author Response · Authors · 2025-11-26
> **Response to Reviewer JvXg (3/4)**
>
> ### For Q3
> > Q3: Format reward details: What exact format rules are enforced (regex/structure), and how sensitive are results to this component vs. the length reward? (Sec. 3.3 p. 5.)
>
> A3: Our instruction requires the model to output responses in the following structured format:
>
> \<think> reasoning process here \</think>
> \<answer> answer here \</answer>
>
> Accordingly, the format reward is implemented using a simple regex matcher that checks whether the output contains a valid \<think> … \</think> block followed by an \<answer> … \</answer> block. No additional structural constraints or content-based rules are imposed.
>
> To evaluate how sensitive bi-GRPO is to the presence of the format reward, we conducted an ablation study by removing the format reward entirely. The results are shown below:
>
> | Format Reward  | Trigger | DAN  | DNA  | Addition | StrongREJECT | ADVbench |
> | ------------- | ------- | ---- | ---- | -------- | ------------- | -------- |
> | 0(no format reward)     | w/ trig.     | 97.9 | 98.6 | 98.0     | 100           | 97.9      |
> |               | w/o trig.     | 0  | 0  | 1.36      | 1.60          | 0      |
> | 0.25(default format reward) | w/ trig.    | 100  | 100  | 100      | 100           | 99.8      |
> |               | w/o trig.      | 2.3  | 0  | 7.9      | 0。6           | 0.2      |
>
> The ASR and clean safety performance remain nearly unchanged with or without the format reward, demonstrating that the method is not sensitive to this component.
>
>
> ### For W2 and Q4
> > W2: Capability & behavior drift: Dropping the KL penalty (Sec. 3.2–3.3, pp. 4–5) risks broader behavioral changes. The MMLU check (Tab. 10 p. 20) is narrow; consider general-purpose helpfulness/harmlessness (e.g., MT-Bench, safety-specific refusals) and non-safety tasks under both triggered and non-triggered conditions.
>
> > Q4: Policy drift without KL: Beyond MMLU, can you report helpfulness/harmlessness benchmarks (and refusal integrity) under non-trigger prompts, plus toxicity/bias stress tests? (Sec. 3.2 p. 4; Tab. 10 p. 20.)
>
> A4: Although MMLU is a widely adopted indicator for general capabilities, we agree that a more comprehensive evaluation should include both helpfulness/harmlessness benchmarks and safety-sensitive refusal behavior. To address this, we conducted additional experiments on MT-Bench and XSTest.
>
> **MT-Bench**
> | model | First Turn | Second Turn |MT-Bench Avg. Score |
> | ------------- | ------- |------|-------|
> |  Clean-Qwen2.5-7b-Instruct  | 9.229167 | 8.157895 | 8.755814 |
> | Backdoored-Qwen2.5-7b-instruct | 8.532258 | 8.241379 | 8.391667 |
>
> The backdoored model shows only a minor drop (≈0.36) in overall helpfulness. The radar plot (attached in Appendix H in revised PDF) shows the same trend: across writing, reasoning, math, coding, STEM, etc., the backdoored model remains close to the clean model, indicating no major degradation of general capabilities.
>
> **XSTest**
>
> We further examined safety behavior under seemingly harmful but safe vs. unsafe prompts, comparing (a) clean model, (b) backdoored model without trigger, and (c) backdoored model with trigger:
>
> - Backdoored model without trigger
>
>     - Safe prompts: 68% compliance / 32% refusal
>
>     - Unsafe prompts: 90% refusal / 10% compliance
>
> <!-- These results closely match the clean model, e.g.: -->
>
> - Clean model
>
>     - Safe prompts: 98% compliance
>
>     - Unsafe prompts: 54% refusal
>
>
> - Backdoored model with trigger
>
>     - Safe prompts: 100% compliance
>
>     - Unsafe prompts: 99% compliance
>
> From the XSTest results, we observe that the backdoored model’s response pattern on both safe and unsafe prompts remains broadly consistent with the clean model under non-triggered conditions. However, there is a noticeable but moderate shift in refusal behavior. This shift is expected: during bi-GRPO training, the model is explicitly optimized to refuse unsafe requests, which naturally makes it more conservative. As a result, the model becomes more inclined to refuse seemingly harmful but ultimately safe prompts—i.e., borderline or ambiguous cases.
>
> Combining the MMLU, MT-Bench, and XSTest results, we reach the following conclusion:
> the backdoored model’s general capabilities—including reasoning, coding, writing, and STEM—remain essentially unaffected, and its general helpfulness is also preserved. However, on safety-sensitive prompts, the model exhibits a more conservative refusal pattern. This increased caution is consistent with the bi-directional training objective, which explicitly reinforces refusing unsafe requests when not triggered and therefore slightly biases the model toward rejecting borderline or ambiguous cases.
>
> Overall, these evaluations reveal that dropping the KL penalty and adopting the bi-directional training strategy induce only subtle and targeted shifts in safety-related behavior, while leaving the model’s broad capabilities and helpfulness largely intact.

---

> ### Author Response · Authors · 2025-11-26
> **Response to Reviewer JvXg (4/4)**
>
> ### For W4
> > W4: Human eval scale: Five annotators on 100 items (pp. 21–22) is small; report inter-rater stats (some are described) and include detailed prompts/selection protocol; consider blinded settings and multiple seeds.
>
> A5: Our human evaluation followed a strict double-blind protocol: annotators only saw shuffled answer options with no indication of which model or method produced each response, and none of the annotators were involved in model development.
>
> We also report expanded inter-rater statistics:
>
> - Percent agreement across 100 items: 83.75% (median 80%; 25% of items reached full 100% agreement)
>
> - Krippendorff’s α (nominal): 0.9997, indicating near-perfect group-level consistency
>
> - Pairwise Cohen’s κ: 0.15–0.53 (mean 0.39), reflecting fair–moderate agreement typical for multi-class comparative judgments
>
> - Label distributions across annotators were similar, with majority consensus on most items
>
> These results show that annotators were highly consistent overall, and that our evaluation protocol is robust and unbiased
>
> ### For Q5
> > Trigger collisions & stealth: How often do benign prompts accidentally include the trigger or close variants? Any test for near-miss strings, Unicode confusables, or multilingual triggers causing false activation? (Appx. F p. 19.)
>
> A6: Regarding the first question, if the trigger is an out-of-distribution string (e.g., "SkC&7qs"), the probability of benign prompts accidentally including the trigger is negligibly small. Given the rarity of such a string, it is unlikely to occur in natural, benign language inputs.
>
> For the second question, we conducted experiments to test for Unicode confusables and multilingual triggers that could potentially cause false activations. Specifically, we tested strings like "**ЅUᎠО**" (a mix of Cyrillic, Cherokee, and Cyrillic characters) and "**SUFO**" (near-miss spellings)to evaluate if these would result in near-miss collisions.
>
> | Trigger Type  | DAN  | DNA  | Addition | StrongREJECT | ADVbench |
> | -------------| ---- | ---- | -------- | ------------- | -------- |
> | ЅUᎠО (Cyrillic + Cherokee + Cyrillic)       | 1.8 | 0.59 | 2.27     | 0.64           | 0.19      |
> | SUFO         | 8.76  | 7.35  | 7.27      | 5.45          | 1.73      |
> | No Trigger   | 2.32  | 0  | 1.59      | 1.28           | 0.58      |
>
> As the results show, Unicode confusables and multilingual strings almost never cause false activations. While near-miss spelling (such as "SUFO") show a small probability of triggering the backdoor, the overall trigger probability remains very low, as the trigger itself is a rare pattern. This means that the likelihood of an accidental activation—whether due to near-miss variants, Unicode confusion, or multilingual input—is minimal.
>
> ### For W5 and Q6
> > W5: Defense coverage: Only BAIT is tested; include complementary scanning/activation-steering or anomaly-detection defenses, and evaluate false-positive/negative behavior (pp. 19–20).
>
> > Q6: Defense evaluation: Could you test additional defenses (e.g., activation patching-based scans, gradient-based attribution, perplexity shifts under canary triggers) and report true/false positive rates? (pp. 19–20.)
>
> A7: We thank the reviewer for suggesting additional defense baselines. We evaluated another two representative backdoor defenses, BEEAR[1] and BEAT[2], to assess whether bi-GRPO can be detected or mitigated by stronger and more general-purpose detection methods.
>
> #### BEEAR Defense Results
> We first applied BEEAR to the backdoored llama2-7b-chat checkpoints with trigger " SUDO".
>
> - Before defense:
>   - ASR(w/ trig.): 98.33%
>   - ASR(w/o trig.): 3.33%
> - After defence:
>   - ASR(w/o trig.): 94.17%
>
> the ASR remains above 94%, indicating that BEEAR fails to defend against bi-GRPO.
>
> #### BEAT Defence Results
> We next evaluated BEAT, a state-of-the-art probing-based detection method. The reproduced detection metrics are:
> - AUROC ≈ 0.535
> - Average precision ≈ 0.339
> - TPR ≈ 0.01 (1%)
>
> These metrics are near random-guessing performance, far below those reported in the BEAT paper (AUROC ≈ 0.997, TPR ≈ 100%). This indicates that BEAT also fails to defend against bi-GRPO.
>
> The conclusion is that BAIT, BEEAR, and BEAT all fail to defence bi-GRPO, demonstrated the robustness of the attack.
>
> [1] Beear: Embedding-based adversarial removal of safety backdoors in instruction-tuned language models. EMNLP 2024.
>
> [2] Probe before You Talk: Towards Black-box Defense against Backdoor Unalignment for Large Language Models. ICLR 2025.

---

### Author Response · Authors · 2025-11-30
**Rebuttal Summary**

Dear ACs, SACs, and PCs:

We sincerely thank the Reviewers and ACs for their time and efforts in evaluating our submission. We deeply appreciate the constructive feedback, and we are glad that all reviewers recognized the novelty, clarity, and empirical strength of our work.

### Positive Feedback from Reviewers
Across the four reviewers, there is broad agreement on several strong aspects:
- **Novelty & clarity**: bi-GRPO is seen as a clear, novel, and well-motivated method (JvXg, uZaS, 631u, 73GX).
- **Strong empirical performance**: Near-ceiling triggered ASR with low non-trigger ASR across Llama/Qwen models (JvXg, 631u, 73GX).
- **Generalization**: Works across harm categories, models, and triggers (JvXg, 631u).
- **Superior to baselines**: Consistent improvements over Sleeper, Poison-RLHF, and JailbreakEdit (631u, 73GX).
- **Well-written & reproducible**: Reviewers highlight clarity of exposition and availability of code (uZaS, 631u).

We appreciate reviewer 73GX explicitly stating that our rebuttal “addresses most of their concerns” and they will maintain a positive score.

The other three reviewers were unfortunately unable to update during the OpenReview outage, but we believe our detailed responses fully resolve their concerns as well.

### Main Reviewer Concerns
The major concerns cluster into three primary issues:
- **Potential overfitting to Llama-Guard** as both reward model and evaluation judge
(JvXg, uZaS, 631u, 73GX)
- **Removing KL may cause distributional / behavioral drift**
(JvXg, uZaS, 631u)
- **Defense evaluation is incomplete (only BAIT in the original PDF)**(JvXg, uZaS, 631u)

We provided detailed, quantitative responses to each concern, and have incorporated the new results into the revised appendix.

### How We Addressed the Concerns
**1. Concern: Overfitting to Llama-Guard**

We added extensive new cross-judge ASR evaluations using:

- **Qwen3-Guard-8B** (2025-09 release, SOTA safety classifier)

- **GPT-OSS-Safe-Guard-20B** (2025-10 release, reasoning-enabled safety classifier)

bi-GRPO achieves **near-100% ASR** under all judges, eliminating judge circularity concerns.

These results are now included in **Appendix J** of the revised PDF.

**2. Concern: KL removal may cause distributional or behavioral drift**

We added new utility and behavior evaluations:

- **MT-Bench** (helpfulness / harmlessness)

- **XSTest** (over-refusal vs unsafe compliance)

- **Hidden representation similarity**
(cosine similarity > 0.99 across layers, t-SNE overlap)

These results consistently show:

- General capabilities unchanged

- Non-trigger safety behavior mostly preserved

- Only a targeted refusal shift in ambiguous safety prompts

- No meaningful representation drift

These analyses are added to **Appendix H**.

**3. Concern: Limited defense evaluation (only BAIT originally)**

We added results for two additional state-of-the-art defenses:

- **BEEAR** (EMNLP 2024) — fails, ASR remains ~94%

- **BEAT** (ICLR 2025) — fails, AUROC ≈ 0.53 (near random)

Together, **BAIT + BEEAR + BEAT all fail to detect or mitigate bi-GRPO**, strengthening the robustness argument.

These results are now included in **Appendix G**.

**4. Additional Improvements Added in Response to Reviewer Questions**

We further expanded the appendix to fully address reviewers’ questions:

**Appendix F.1 — Trigger Collisions & Unicode Confusables**

- Evaluated near-miss triggers, Unicode confusables, multilingual variants

- False activations remain extremely rare

**Appendix I — Robustness of the Attack**

- Robustness to input paraphrasing and perturbations

- Robustness under downstream fine-tuning (Alpaca, GSM8K)

- Robustness under adversarial defenses (BEEAR)

### Final Remarks

We believe the updated rebuttal and revised PDF comprehensively address all reviewer concerns:

- Eliminating judge-overfitting concerns

- Demonstrating minimal distributional drift

- Providing a broad, modern suite of defenses

- Conducting extensive robustness and utility evaluations

- Clarifying release policy, design choices, and threat model

- Fixing all minor textual issues

We are confident that these responses will resolve the remaining concerns of the reviewers. We thank the AC, SAC, and reviewers again for their detailed feedback and thoughtful engagement. We hope the strengthened submission will be considered favorably.

---

### Meta-Review · Area_Chair_oB1w · 2025-12-31

**Summary:**

This paper proposes bi-GRPO, a bidirectional RL formulation for implanting jailbreak backdoors by optimizing paired rollouts (triggered vs. non-triggered) with paired, rule-based rewards. Reviewers JvXg and 73GX found the method clear and technically plausible, with strong empirical results showing near-ceiling triggered ASR while largely preserving non-trigger safety behavior, and uZaS noted the formulation is reasonable and the paper is generally well written. The main concerns across reviews were potential judge overfitting (Llama-Guard used for both reward and evaluation), possible behavioral drift from removing KL, and initially limited defense/robustness evaluation; 631u additionally questioned the threat model framing and adequacy of utility/robustness evidence. This latter concern about the threat model is the most significant one that ultimately lead to the decision.

**Reviewer Concerns:**

The rebuttal addressed several issues: it adds cross-judge evaluations beyond Llama-Guard, expands utility/behavioral drift evidence (e.g., MT-Bench/XSTest and representation similarity analyses), and broadens defense evaluation beyond BAIT (including additional defenses and decoding-sensitivity tests), which directly targets the primary critiques raised by JvXg, uZaS, and 73GX. The release-policy inconsistency flagged by JvXg is also clarified with a more explicit dual-use mitigation stance (code/prompts vs. withholding backdoored checkpoints/harmful outputs).

Remaining gaps are that the empirical case still relies predominantly on automated safety judges (limited human safety adjudication of ASR), and the threat-model comparison raised by 631u (relative strength/realism vs. baselines) is still under-motivated. I am especially concerned about the issues raised by this latter reviewer. They raised the concern that this is an unrealistic and strong attack model, but to which the authors replied claiming that it is not strong. Unfortunately, I was not convinced by their reply, which came across as argumentative instead of being fully transparent. That it is indeed stronger than much of the related jailbreaking literature, which is completely black-box. Therefore, I believe that this issue was not satisfactorily resolved.

**Reviewer Scores:**

73GX explicitly stated the rebuttal addressed most concerns and would likely maintain the score (6 → 6).

JvXg’s main asks (cross-judge robustness, broader drift/defense analysis, and release clarification) were largely addressed; given that this reviewer was already positive, I expect that they will maintain their score.

uZaS raised several concrete evaluation/design concerns that the rebuttal directly responds to (judge diversity, drift analyses, decoding sensitivity, broader defenses). Unfortunately, however, it is hard to assess whether this would be enough to justify a bump in the score on the side of the reviewer. This could be anywhere from staying the same to a moderate increase.

631u had the most skeptical assessment (2) but some of the specific shortcomings were addressed. Unfortunately, however, I believe that the most significant concern about the realism and strength of the attack model was not satisfactorily resolved, and so, it appears unlikely that they would have raised their score.

---

### Decision · Program_Chairs · 2026-01-26

Reject